# Observation of photonic anomalous Floquet topological insulators

Lukas J. Maczewsky[1,*], Julia M. Zeuner[1,*], Stefan Nolte[1] & Alexander Szameit[1]

Topological insulators are a new class of materials that exhibit robust and scatter-free transport along their edges — independently of the fine details of the system and of the edge — due to topological protection. To classify the topological character of two-dimensional systems without additional symmetries, one commonly uses Chern numbers, as their sum computed from all bands below a specific bandgap is equal to the net number of chiral edge modes traversing this gap. However, this is strictly valid only in settings with static Hamiltonians. The Chern numbers do not give a full characterization of the topological properties of periodically driven systems. In our work, we implement a system where chiral edge modes exist although the Chern numbers of all bands are zero. We employ periodically driven photonic waveguide lattices and demonstrate topologically protected scatter-free edge transport in such anomalous Floquet topological insulators.

[1] Institute of Applied Physics, Abbe Center of Photonics, Friedrich-Schiller-Universität Jena, Max-Wien-Platz 1, 07743 Jena, Germany. * These authors contributed equally to this work. Correspondence and requests for materials should be addressed to A.S. (email: alexander.szameit@uni-jena.de).

The discovery of the quantized Hall effect[1] revealed the existence of a new class of extremely robust transport phenomena, which are largely independent of sample size, shape and composition. The scatter-free nature of these phenomena can be linked to the existence of non-trivial topological invariants associated with the systems' bulk bands[2]. Shortly after the discovery of topological insulators[2–7], the concept of topology was transferred to the photonic domain of electromagnetic waves[8] with the first realization in the microwave regime implementing the photonic analogue of the quantum Hall effect[9]. The search for an optical realization of topological insulators has prompted a number of proposals[10–14], and culminated in various experimental realizations[5,6]. Photonic topological insulators may enable novel and more robust photonic devices such as waveguides, interconnects, delay lines, isolators and couplers (or anything susceptible to parasitic scattering by fabrication disorder). The field of topological photonics[15] evolved well afterwards and resulted in various further studies, such as nonlinear waves in topological insulators and the prediction of topological gap solitons[16], topological states in passive PT-symmetric media[17], topological sub-wavelength

**Figure 1 | Floquet band structure in a driven system.** Conceptual sketch of the band structure in a driven system, which is periodic in momentum $k$ and quasi-energy $\varepsilon$. Essentially, the band structure is analogue to a torus (see inset). This allows chiral edge modes to exist even if the Chern numbers of all bands are equal to zero.

**Figure 2 | Bipartite lattice structure with periodic driving.** (**a**) The coupling to the neighbouring waveguides occurs in four steps of equal length; in each step, hopping takes place solely along the highlighted bonds with a coupling strength $c_j$; all other couplings are zero. (**b**) If the coupling during each step is 100% $\left(c_j = \frac{2\pi}{T}\right)$, after a full driving period $T$, one observes the formation of localized bulk modes without dispersion and chiral edge modes travelling along the lattice boundaries. (**c**) A schematic sketch of four lattice sites of the fabricated sample, in which the waveguides are drawn pairwise together to enable evanescent coupling. The initial waveguide spacing is $a = 40\,\mu m$ ensuring negligible coupling between adjacent guides. (**d**) The edge band structure of periodic quasi-energies in the case $c_j = \frac{2\pi}{T}$, exhibiting a flat bulk band (brown line) and dispersionless chiral edge modes. Dotted and solid orange lines describe the dispersion of opposite edges, respectively.

settings[18] and even three-dimensional systems exhibiting Weyl points[19].

It is commonly accepted that for two-dimensional spin-decoupled topological systems a complete topological characterization is provided by the Chern numbers of each band, which represent a set of integer topological invariants[20,21]. The number of chiral edge modes residing in a bandgap is given by the sum of the Chern numbers of all bands below this gap. Hence, the Chern number is equal to the difference between the chiral edge modes entering the band from below and exiting it above[15]. However, this is strictly true only for systems that are static, that is, where the Hamiltonian is constant in time. In periodically driven (Floquet) systems, the Chern numbers employed in the static case do not give a full characterization of the topological properties[22]. The reason is that in these systems, the fixed energy in the band structure is replaced by a periodic quasi-energy. As a consequence, the Chern numbers of all bands lying below a certain gap cannot be summed up since there exists no lowest band in the (periodic) band structure. Moreover, in such systems chiral edge modes are possible[10,23], although the Chern numbers of all bands may be zero (see Fig. 1 for an illustrative sketch). These materials are called anomalous Floquet topological insulators (A-FTI)[22,24]. Recently, it was shown that the appropriate topological invariants for characterizing these new phenomena are winding numbers[22], which utilize the information in the Hamiltonian for all times within a single driving period. This is in contrast to the Chern numbers of the individual bands, which only depend on the Hamiltonian evaluated stroboscopically once per driving cycle. Recently, anomalous edge states were shown in static network systems that are described by a scattering matrix and can be mapped onto a Floquet lattice[25,26]. However, to date the experimental demonstration of an A-FTI in an explicitly driven system is still elusive.

## Results

### Tight-binding and Floquet description of the lattice.

In our work, we experimentally demonstrate an A-FTI in a two-dimensional driven system being not only periodical in the lattice directions $x$ and $y$ but also along the evolution coordinate. To this end, we work in the photonic regime and employ arrays of evanescently coupled waveguides. In such structures, the light evolution is governed by the paraxial Helmholtz equation, which is mathematically equivalent to the Schrödinger equation (see ref. 27 for details). Therefore, evanescently coupled waveguide lattices are an excellent platform for testing Schrödinger physics.

We consider a bipartite square lattice with two site species $A$ and $B$ (with same on-site potential), as it was suggested in ref. 22. Along the propagation direction, the structure consists of four sections with each having length $T/4$ and the entire period is $T$. In the first section, a particular A-site couples to neighbouring B-site on its right, in the second section to its neighbouring B-site above, and in the third and fourth section to its left and below, respectively (as sketched in Fig. 2a). If a 100%-coupling per section $\left(c_j = \frac{2\pi}{T}\right)$ is present, this lattice structure exhibits no transport in the bulk, as an excitation is trapped by moving only in loops, whereas at the edge transport occurs (see Fig. 2b). Figure 2c shows a sketch of how we realized this lattice in our experiments. The inter-site coupling in the individual sections $n$ is achieved by appropriately engineering directional couplers[28]. This system is described by the Bloch Hamiltonian

$$H_{\mathrm{B}}(\mathbf{k}, z) = -\sum_{j=1}^{4} \begin{pmatrix} 0 & c_j(z)e^{i\mathbf{b}_j \mathbf{k}} \\ c_j(z)e^{-i\mathbf{b}_j \mathbf{k}} & 0 \end{pmatrix},$$

where the vectors $\{\mathbf{b}_j\}$ are given by $\mathbf{b}_1 = -\mathbf{b}_3 = (a,0)$ and $\mathbf{b}_2 = -\mathbf{b}_4 = (0,a)$, with $a$ being the distance between adjacent lattice sites. In addition, for each partial step $n$, the coupling coefficients $\{c_j(z)\}$ are defined as $c_j = \delta_{jn}c$. We start our analysis by choosing the coupling coefficient $c = \frac{2\pi}{T}$, such that during each step complete coupling into the respective neighbouring waveguide occurs. Obviously, the Hamiltonian is $z$-dependent, which for waveguide lattices is analogue to time-dependence in

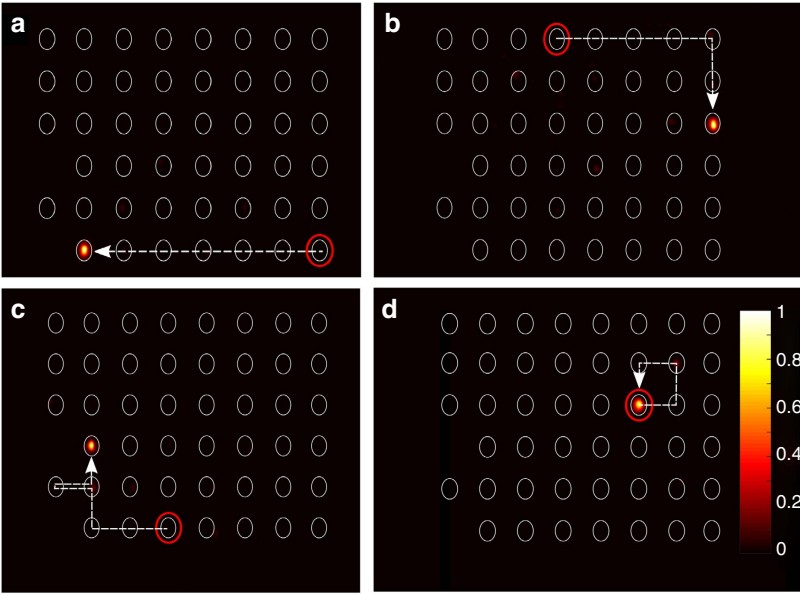

**Figure 3 | Experimental demonstration of flat band structure.** Light distribution in the lattice after single-waveguide excitation for perfect coupling with strength $c_j = \frac{2\pi}{T} = \frac{2\pi}{40}$mm$^{-1}$ and after 3 full periods. The waveguide positions of the lattice are marked by white ellipses, the excited site is marked by a red ellipse, and the trajectory is visualized with a white arrow. Evolution of the excited chiral edge state (**a**) along the edge, (**b**) around a corner and (**c**) along artificial defects in the lattice structure. (**d**) If a bulk waveguide is excited, light follows a loop trajectory, as only localized flat band modes are excited. The intensity in each figure is normalized to its maximum in the figure. One can clearly see that no scattering and no dispersion occurs, supporting the claim that a dispersion-free chiral edge state was excited.

quantum mechanics[27] and, hence, no eigenstates exist. However, due to the periodicity in $z$, Floquet theory can be applied to derive a band structure of so-called quasi-energies $\varepsilon$ (ref. 22). A solution of such a time-dependent Schrödinger equation are the Floquet states $\psi(t) = \phi(t)e^{-i\varepsilon t}$ with $\phi(t+T) = \phi(t)$. Consequently, the Floquet spectrum is periodic in its quasi-energies, in full correspondence to the periodicity in the transverse momentum caused by Bloch's theorem. The temporal evolution of the system is described by $\psi(t) = Pe^{-i\int_0^t H(\tau)d\tau}\psi(0)$, such that $\psi(T) = e^{-i\varepsilon T}\psi(0)$. Note that $P$ is the time-ordering operator. The time evolution operator $U(t) = Pe^{-i\int_0^t H(\tau)d\tau}$ includes the effective stroboscopic dynamics after multiples of the period $T$ and the micro motion within a single period. This represents the full Floquet regime, in contrast to the adiabatic system used in our recent work[5], in which the high-frequency driving allows for the description with an effective time-independent Hamiltonian $H_{\text{eff}}$ for all times $t$ as $\psi(t) = e^{-itH_{\text{eff}}}\psi(0)$.

**Topological characterization by winding number.** Our lattice structure exhibits two flat degenerate bands (that appear as a single band), as the bipartite character of the lattice arises only from the sequential coupling steps with four equal coupling coefficients $c_j$ and not from a sublattice potential. Since the sum of the Chern numbers of all bands has to be zero, we find that the Chern number of the flat band in our system is zero. Although, when considering a finite system, we observe the formation of chiral edge states (see Fig. 2d). In this vein, the Chern number is not the appropriate topological invariant that characterizes the existence and the amount of chiral edge states in our system. This is the very nature of an A-FTI. As it was shown earlier[22], in periodically driven systems, the topological invariant characterizing the number of chiral edge modes is the winding number $W_\varepsilon$, which is equal to the number of chiral edge modes $n_{\text{edge}}$ in a bandgap at a certain quasi-energy $\varepsilon$:

$$n_{\text{edge}}(\varepsilon) = W_\varepsilon.$$

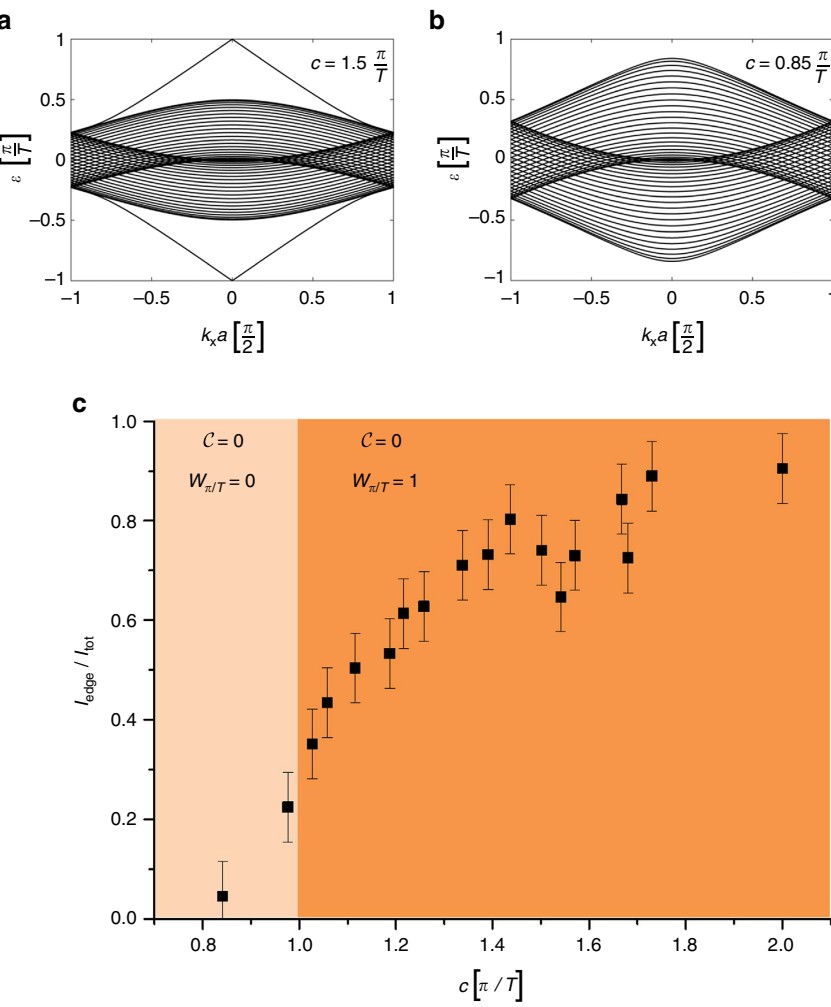

**Figure 4 | Winding number transition.** (**a**) If the coupling coefficients are reduced to $c = 1.5\frac{\pi}{T}$, the bulk band is not flat anymore. However, chiral edge modes still exist in the center of the Brillouin zone, and the winding number remains $W_{\pi/T} = 1$. (**b**) For $c = 0.85\frac{\pi}{T}$ the chiral edge states have disappeared, and the winding number is $W_{\pi/T} = 0$. (**c**) Visualization of the phase transition using a single-waveguide excitation. As this populates all modes, the fraction of the trapped intensity at the surface $I_{\text{edge}}$ with respect to the total intensity $I_{\text{tot}}$ as a function of the coupling strength indicates the amount of existing chiral edge modes. When the coupling is $c < \frac{\pi}{T}$, these modes disappear and essentially all light diffracts into the bulk. The different topological phases are marked with dark orange ($W_{\pi/T} = 1$) and light orange ($W_{\pi/T} = 0$), in both regimes, however, the Chern number $\mathcal{C}$ is zero. The error bars result due to uncertainties of the fabrication process and are estimated via linear propagation of errors.

The winding number is directly related to the Chern number[22]:

$$W_{\varepsilon_2} - W_{\varepsilon_1} = \mathcal{C}_{\varepsilon_1 \varepsilon_2},$$

where $\mathcal{C}_{\varepsilon_1 \varepsilon_2}$ is the sum of the Chern numbers of all bands residing between $\varepsilon_1$ and $\varepsilon_2$. Therefore, the difference of the number of chiral edge modes entering a band from below and exiting it above is equal to the Chern number of the respective band. We describe the approach for calculating the winding number in Methods section.

**Experimental realization of flat band structure.** For our experiments, we fabricate the lattice sketched in Fig. 2c using the laser direct-writing technology[27]. For details regarding the fabrication, the lattice parameters and the characterization setup we refer to Methods section. We start by launching light into single sites of the lattice and observe light dynamics that is summarized in Fig. 3. As clearly shown, the excited edge state travels dispersionless and without any scattering along edges, around corners and various defects (Fig. 3a–c). This highly robust, unidirectional state is a clear signature of topological protection. However, as opposed to a common Floquet topological insulator, in our system we find a flat band of bulk modes. This is shown by exciting the sites in the bulk of the lattice and observing that light is trapped in a loop, indicating the excitation of only localized modes (see Fig. 3d for one example). As we observe the same dynamics for any bulk site, we can conclude that there is indeed only one band, which consists of localized degenerate states: a single flat band, which has to have a Chern number of zero. This is the unequivocal proof of having implemented an A-FTI, as clearly the Chern number does not predict the existence of the chiral edge states.

**Examination of the winding number transition.** In the next step, we will analyse the impact of the inter-site coupling on the topological nature of the system. So far we considered perfect hopping $\left(c = \frac{2\pi}{T}\right)$, that is, in each section $n$ the light completely couples to the neighbouring site, which results in an A-FTI phase. However, when decreasing the hopping rate (which results in only partial coupling), one will eventually leave the topologically non-trivial phase[22] and enter the trivial phase exactly at $c = \frac{\pi}{T}$. This is clearly visible in the edge band structures: one example of the topological non-trivial regime is shown in Fig. 4a $\left(c = 1.5\frac{\pi}{T}\right)$ and an example of the trivial regime in Fig. 4b $\left(c = 0.85\frac{\pi}{T}\right)$. Whereas for $c > \frac{\pi}{T}$ chiral edge states exist (topological phase, Fig. 4a), at $c = \frac{\pi}{T}$ a phase transition occurs and the edge states disappear, such that for $c < \frac{\pi}{T}$ the system is in a trivial phase (Fig. 4b). Note, that in both phases the Chern number of the band is zero, and only the value of the winding number changes. To study this phase transition, we perform various measurements in systems with decreasing coupling constant (see Methods section for the experimental approach). We launch light into a single site at the edge of the structure, as this populates the entire band structure, and analyse the diffraction pattern. If there is an edge state present, it is partially excited by the single-site excitation and some of the evolving light will remain at the edge during propagation. However, if there is no edge state present, after a certain propagation length all of the light will have diffracted into the bulk of the system. Our experimental results are summarized in Fig. 4c, where we plot the intensity ratio $I_{\text{edge}}/I_{\text{tot}}$ as a function of the coupling constant $c$. The error bars are due to slightly fluctuating power of the writing laser and the signal to noise ratio of the recorded charge-coupled device (CCD) images. For $c = \frac{2\pi}{T}$ indeed almost all of the light remains at the edge, as suggested by the edge band structure shown in Fig. 2d. For a decreasing

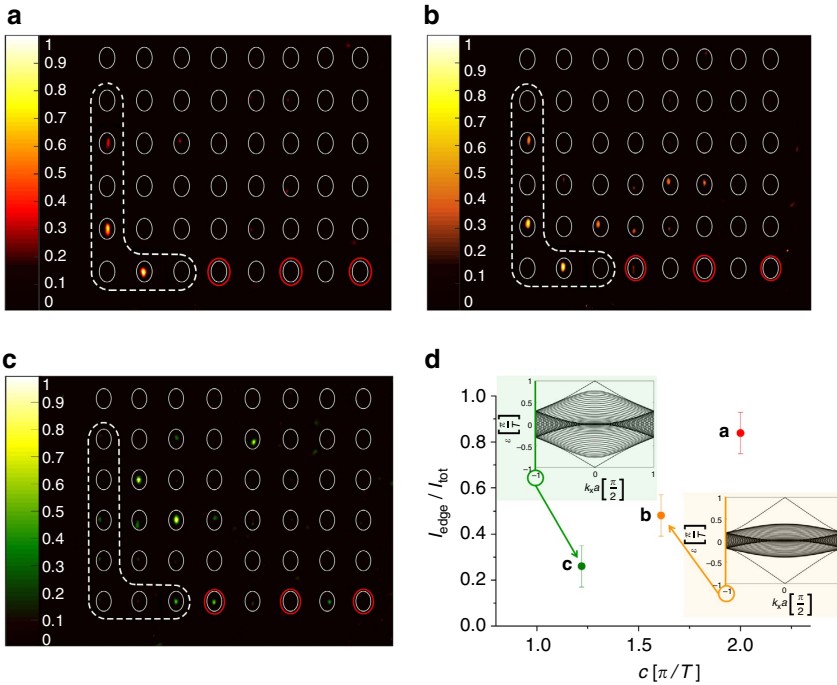

**Figure 5 | Edge state observation for specific momentum excitation.** The region where separated chiral edge modes exist in the edge band structure reduces with decreasing coupling strength $c$. This is shown by exciting the band structure at $k_x = -\frac{\pi}{2a}$ using an appropriately tilted broad beam and observing the diffraction pattern for (**a**) $c = \frac{2\pi}{T}$, (**b**) $c = 1.63\frac{\pi}{T}$ and (**c**) $c = 1.2\frac{\pi}{T}$. The excited waveguides are marked by red ellipses, the integration area to calculate the part of the intensity residing at the edge $I_{\text{edge}}$ is surrounded by a white dashed line. (**d**) The plot of the intensity fraction of the trapped light clearly indicates that at $k_x = -\frac{\pi}{2a}$ the amount of intensity exciting edge states significantly reduces for decreasing coupling strength. The intensity in each figure is normalized to its maximum. The respective edge band structures for $c = 1.63\frac{\pi}{T}$ and $c = 1.2\frac{\pi}{T}$ are shown as insets. The error bars result due to uncertainties of the fabrication process and are estimated via linear propagation of errors.

coupling constant the fraction of light that remains at the edge monotonously decreases until at $c < \frac{\pi}{T}$ no edge states are present, as the trivial phase is reached.

### Edge state observation for specific momentum excitation.

Importantly, the region in reciprocal space where chiral edge states being separated from the bulk bands are found reduces for decreasing coupling strength: whereas in the centre of the edge band structure around $k_x = 0$ such states are always found until the phase transition occurs, separated edge modes close to the edge of the band structure (around $k_x = -\frac{\pi}{2a}$) continuously cease to exist for decreasing coupling. This is illustrated when exciting the band structure only at the specific momentum $k_x = -\frac{\pi}{2a}$ by using an appropriately tilted broad beam[29]. In Fig. 5a–c, it is shown that the state completely remains at the edge of the lattice for $c = 2\frac{\pi}{T}$ (Fig. 5a), and partially spreads into the bulk for $c = 1.63\frac{\pi}{T}$ while a significant fraction is still trapped at the edge (Fig. 5b). However, for $c = 1.2\frac{\pi}{T}$ the light almost completely diffracts away from the edge as no separated chiral edge modes remain at $k_x = -\frac{\pi}{2a}$ for this low coupling strength (Fig. 5c). Note that the chiral edge states reside on every second waveguide solely, such that we excited only those with the broad beam. Our results are summarized in Fig. 5d, where the fraction of the light trapped at the lattice edge is plotted as a function of the coupling strength. One clearly sees the drop in light intensity at the edge, proving the disappearance of the edge states for decreasing coupling strength. In addition, the edge band structures for $c = 1.63\frac{\pi}{T}$ and $c = 1.2\frac{\pi}{T}$ are shown as insets to see the region in k-space in which the topological edge states exist. For $c = 2\frac{\pi}{T}$, the respective edge band structure is equal to Fig. 2d.

### Discussion

Summarizing our work, the results presented here clearly demonstrate the significance of the winding number as the appropriate topological invariant characterizing periodically driven systems. Moreover, the chiral edge states in A-FTIs are highly robust to distortions in the lattice structure (including defects and imperfect hopping). Hence, our experimental observation of an A-FTI opens a new chapter in the field of topological physics. Only recently, a novel topological phase was predicted in disordered A-FTI: the anomalous Floquet-Anderson insulator[30]. But there are many more puzzles to solve: What is the impact of nonlinearity on the formation of these chiral edge states? Does the dimensionality play a significant role? What are the possibilities to obtain different phases than reported here? The answer to these and other intriguing questions are now in reach. The authors of this work would like to point out that a related work with similar results is published in ref. 32.

### Methods

**Winding number.** To calculate the number of chiral edge modes in a periodically driven system, the behaviour of the system during a full driving period has to be taken into account, by employing the time evolution operator $U(\mathbf{k}, t) = P \exp\left(-i \int_0^t dt' H(\mathbf{k}, t')\right)$, with $\mathbf{k}$ as the momentum and $P$ as the time-ordering operator. In a system exhibiting a flat band at quasi-energy $\varepsilon = 0$, the winding number $W$ can be calculated as[22]:

$$n_{\text{edge}} = W[U] = \frac{1}{8\pi^2} \int dt dk_x dk_y \cdot Tr\left(U^{-1}\partial_t U \cdot \left[U^{-1}\partial_{k_x} U, U^{-1}\partial_{k_y} U\right]\right).$$

In the case of curved (dispersive) bands, the winding number in a gap is $W_\varepsilon = W[U_\varepsilon]$, with $U_\varepsilon$ being constructed as follows[22]:

$$U_\varepsilon(\mathbf{k}, t) = \begin{cases} U(\mathbf{k}, 2t) & \text{if } 0 \leq t \leq \frac{T}{2} \\ V_\varepsilon(\mathbf{k}, 2T - 2t) & \text{if } \frac{T}{2} \leq t \leq T. \end{cases}$$

Here, $V_\varepsilon(\mathbf{k}, t) = \exp(-iH_{\text{eff}}(\mathbf{k})t)$ with $H_{\text{eff}}(\mathbf{k}) = \frac{i}{T}\log U(\mathbf{k}, T)$. The branch cut of

the logarithm is chosen such that:

$$\log e^{-i\varepsilon T + i0^-} = -i\varepsilon T,$$
$$\log e^{-i\varepsilon T + i0^+} = -i\varepsilon T - 2\pi i.$$

**Fabrication of the structures.** The single-mode waveguides were written[27] inside a high-purity 15-cm-long fused silica wafer (Corning 7980) using a RegA 9000 seeded by a Mira Ti:Al$_2$O$_3$ femtosecond laser. Pulses centred at 800 nm with duration of 150 fs were used at a repetition rate of 100 kHz and energy of 450 nJ. The pulses were focused 671 to 883 μm under the sample surface using an objective with a numerical aperture (NA) of 0.35, while the sample was translated at constant speed of 100 mm min$^{-1}$ by high-precision positioning stages (ALS130, Aerotech Inc.). The refractive index increase of each guide is $\sim 8 \times 10^{-4}$, the mode field diameters of the guided mode were 10.4 μm × 8.0 μm at 633 nm. Propagation losses and birefringence were estimated at 0.2 dB cm$^{-1}$ and in the order of $10^{-7}$, respectively. The site spacing $a = 40$ μm ensures that there is no unwanted coupling between adjacent waveguides. In the individual sections of the lattice (shown in Fig. 2c) the waveguides that couple converge to a spacing of 9.7 μm to ensure significant inter-site hopping. For perfect coupling $\left(c_j = 2\frac{\pi}{T}\right)$, the length of a full period is $T = 40$ mm. Each bending is 4.17 mm long, such that the additional losses caused by the bending are as low as 4%. The coupling strength between the guides was determined in preliminary experiments as a function of inter-site spacing and interaction length; experimental errors arising due to uncertainties in the fabrication and the measurement are $\sim 5\%$. For obtaining the different coupling strengths between the individual sites the length of the coupling region was appropriately designed, taking into account the weak coupling that occurs already in the bends[28]. All samples contain 3 full periods, whereas the remaining 3 cm were used for preparation of the injection distribution required in each case.

**Characterization of the structures.** For the observation of the light evolution, light from a tunable Helium Neon laser (Thorlabs HTPS-EC-1) was launched into the system using a NA = 0.35 objective. Whereas this is sufficient for single-site excitation, for the broad excitation the beam was expanded with a slit and a biconvex lens ($f = 35$ mm) perpendicular to the orientation of the slit. Together with every other waveguide starting 2 cm later in propagation direction and an appropriate tilt of the sample we excite the correct transverse momentum[29]. We fabricated several structures with different coupling strengths as described above. However, in order to achieve more data points, we used different excitation wavelengths (633, 604, 594 and 543 nm) that allowed us to further manipulate the coupling strength[31]. We calibrated the wavelength-dependent coupling strength for the different interaction lengths of the individual sections in independent directional couplers.

**Data availability.** The data that support the findings of this study are available from the corresponding author (A.S.) upon reasonable request.

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

## Acknowledgements

We gratefully acknowledge financial support from the Deutsche Forschungsgemeinschaft (grants SZ 276/7-1, SZ 276/9-1, BL 574/13-1, GRK 2101/1) and the German Ministry for Science and Education (grant 03Z1HN31).

## Author contributions

L.J.M. performed the measurements, J.M.Z. elaborated on the theory and A.S. supervised the project. All authors discussed the results and co-wrote the paper.

## Additional information

**Competing financial interests:** The authors declare no competing financial interests.

