## [Peer Review File · Nature Communications]

REVIEWERS' COMMENTS:

Reviewer #1 (Remarks to the Author):

The authors present an all-optical realization of a topological phase which is unique to time dependent systems with a periodic time dependence. In their realization, a "time-dependent" tight bonding system is realized by an array of coupled wave guides, and the time evolution corresponds to propagation along the propagation axis of the light, following the theoretical proposal discussed in Physical Review X 3, 244 031005 (2013). Topological Floquet band-structures have been realized previously in optical systems, for example in Nature 496, 196 259 (2013), however these previous experiments realized time-dependent systems whose topological properties can be captured by considering an evolution with a static "effective" Hamiltonian. Therefore, these topological phases are direct analogues to those realized in static systems. The novelty of the current experiment is that it realizes a system whose properties cannot be realized in a static system. The results presented by the authors give convincing evidence that indeed they succeeded to prepare the waveguide system in the anomalous topological phase. This represents a major advancement in the field of topological, time dependent systems that has been gaining a lot of attention lately. I therefore recommend that the paper be published in Nature Communications.

A small note: The authors refer to the approximation of an evolution with an "effective Hamiltonian" as an "adiabatic approximation". This seem to me to be an incorrect terminology.

The authors have corrected all of the comments I gave in my review of their manuscript for Nature Physics in a satisfactory manner.

Reviewer #2 (Remarks to the Author):

1) Based on the authors' response below, it doesn't seem that the authors have a clear understanding of the literature.

Scattering matrix is just a method of choice to construct the eigenvalue or Hamiltonian problem. See the following paper describing a similar system as those in Ref. [25] and [26]. Hafezi M, Demler E A, Lukin M D, et al. Robust optical delay lines with topological protection[J]. Nature Physics, 2011, 7(11): 907-912.

In short, the authors's view on Ref. [25] and [26] in their response and in the paper is incorrect.

=====

AUTHOR RESPONSE

=====

The works the referee refers to (references [25] and [26] in our paper) are NOT Hamiltonian systems; they exhibit only a scattering matrix and, hence, do not fulfill the conditions for a topological insulator (which has to have a Hamiltonian). These works show the existence of anomalous edge states, but NOT the implementation of an anomalous topological insulator, which is a very important distinction!

The difference is as important as between, e.g., topological edge states in graphene (which always exist) and topologically protected edge states in graphene (which is hard to do; this is actually the achievement of our recent Nature paper on topological insulators [5]).

IN PAPER:

Recently, anomalous edge states were shown in network systems that are described by a scattering matrix [25,26]. However, to date the experimental demonstration of A-FTI is still elusive.

=====

2) I wonder if "insulator" in the title is a good name for describing such a photonic waveguide array.

1. "Insulator" already has a very clear definition.

2. It is confusing across different communities.

3. It indicates the work is for the purpose of mimicking electronic concepts.

3) In the captions of the figures showing experimental data, it might be helpful to note "experimental" (in the caption) for the ease of recognition.

4) It will be helpful to have an experimental image of the actual lattice array.

REVIEWERS' COMMENTS:

Reviewer #1 (Remarks to the Author):

The authors present an all-optical realization of a topological phase which is unique to time dependent systems with a periodic time dependence. In their realization, a “time-dependent” tight bonding system is realized by an array of coupled wave guides, and the time evolution corresponds to propagation along the propagation axis of the light, following the theoretical proposal discussed in Physical Review X 3, 244 031005 (2013). Topological Floquet band-structures have been realized previously in optical systems, for example in Nature 496, 196 259 (2013), however these previous experiments realized time-dependent systems whose topological properties can be captured by considering an evolution with a static “effective” Hamiltonian. Therefore, these topological phases are direct analogues to those realized in static systems. The novelty of the current experiment is that it realizes a system whose properties cannot be realized in a static system. The results presented by the

authors give convincing evidence that indeed they succeeded to prepare the waveguide system in the anomalous topological phase. This represents a major advancement in the field of topological, time dependent systems that has been gaining a lot of attention lately. I therefore recommend that the paper be published in Nature Communications.

A small note: The authors refer to the approximation of an evolution with an “effective Hamiltonian” as an “adiabatic approximation”. This seem to me to be an incorrect terminology.

RESPONSE:

First, we would like to thank the Reviewer for recommending the paper for publication. She/he is absolutely right, that although in [5] an adiabatic approximation is valid, it should not be mixed up with the effective Hamiltonian description. We changed the text in the manuscript accordingly: “This represents the full Floquet regime, in contrast to the adiabatic system used in our recent work [5], in which the high-frequency driving allows for the description with an effective time-independent Hamiltonian H_{eff} for all times t as $\psi(t) = e^{-itH_{\text{eff}}}\psi(0)$.”

The authors have corrected all of the comments I gave in my review of their manuscript for Nature Physics in a satisfactory manner.

Reviewer #2 (Remarks to the Author):

1) Based on the authors’ response below, it doesn't seem that the authors have a clear understanding of the literature.

Scattering matrix is just a method of choice to construct the eigenvalue or Hamiltonian problem. See the following paper describing a similar system as those in Ref. [25] and [26].

Hafezi M, Demler E A, Lukin M D, et al. Robust optical delay lines with topological protection[J].

Nature Physics, 2011, 7(11): 907-912.

In short, the authors's view on Ref. [25] and [26] in their response and in the paper is incorrect.

=====

AUTHOR RESPONSE

=====

The works the referee refers to (references [25] and [26] in our paper) are NOT Hamiltonian systems; they exhibit only a scattering matrix and, hence, do not fulfill the conditions for a topological insulator (which has to have a Hamiltonian). These works show the existence of anomalous edge states, but NOT the implementation of an anomalous topological insulator, which is a very important distinction! The difference is as important as between, e.g., topological edge states in graphene (which always exist) and topologically protected edge states in graphene (which is hard to do; this is actually the achievement of our recent Nature paper on topological insulators [5]).

IN PAPER:

Recently, anomalous edge states were shown in network systems that are described by a scattering matrix [25,26]. However, to date the experimental demonstration of A-FTI is still elusive.

=====

RESPONSE:

We see that network systems described by a scattering matrix can also be seen as Hamiltonian systems. However, they are not driven systems which explicitly show a periodical modulation along the evolution coordinate, but only can be mapped onto a Floquet Hamiltonian. Hence, we changed the related sentences in the manuscript to point out better that the network systems are static systems while the here shown system really is a periodically driven system being directly described by a Floquet Hamiltonian.

"Recently, anomalous edge states were shown in static network systems that are described by a scattering matrix and can be mapped onto a Floquet lattice [24,25]. However, to date the experimental demonstration of an A-FTI in an explicitly driven system is still elusive.

In our work, we experimentally demonstrate an A-FTI in a two-dimensional driven system being not only periodical in the lattice-directions x and y, but also along the evolution coordinate. To this end, we work in the photonic regime and employ arrays of evanescently coupled waveguides."

2) I wonder if "insulator" in the title is a good name for describing such a photonic waveguide array.

1. "Insulator" already has a very clear definition.
2. It is confusing across different communities.
3. It indicates the work is for the purpose of mimicking electronic concepts.

RESPONSE:

Although we have to partially agree with the Reviewer that insulator is indeed a bit confusing in the scope of photonics, we still would like to keep the name here for the following reasons:

- "photonic topological insulator" has become a well-known and standalone term in optics, see e.g. Refs. [5,14,16]

- it is definitely not confusing within the optics community, and to clearly distinguish from a solid-state topological insulator, the descriptive adjective "photonic" is used

3) In the captions of the figures showing experimental data, it might be helpful to note "experimental" (in the caption) for the ease of recognition.

RESPONSE:

Good comment, we changed the figure captions accordingly.

4) It will be helpful to have an experimental image of the actual lattice array.

RESPONSE:

There is only the possibility of having a microscopic image of the front/back facet of the lattice as attached here. Unfortunately, it is not possible to "look into" the glass chip and image the whole lattice geometry along z. As we do not think that one can gain really more insight by looking at the microscopic image of the front facet, we prefer to not put the image into the manuscript. The only information one can get from the microscopic image is the position of the waveguides which can also be seen in the experimental output image of the light distribution, where the position of the waveguides is marked by the white ellipses. To make that clear in the manuscript, we added the following statement in the caption of Fig.3:

"The waveguide positions of the lattice are marked by a white ellipse..."

Output facet of the waveguide lattice used for edge stage observation for a specific momentum excitation (Fig. 5(a) in manuscript) taken by a microscope.